# Correlation between Microstructural Change and Irradiation Hardening Behavior of He-Irradiated V–Cr–Ti Alloys with Low Ti Addition

Ken-ichi Fukumoto [1,*] , Shuichiro Miura [1], Yoshiki Kitamura [1], Ryoya Ishigami [2] and Takuya Nagasaka [3]

1   Research Institute of Nuclear Engineering, University of Fukui, Fukui 914-0055, Japan;
    hiiragium@gmail.com (S.M.); kitamura.y.ap@m.titech.ac.jp (Y.K.)
2   The Wakasa Wan Energy Research Center, Fukui 914-0135, Japan; rishigami@werc.or.jp
3   National Institute of Fusion Science, Gifu 509-5202, Japan; nagasaka.takuya@nifs.ac.jp
*   Correspondence: fukumoto@u-fukui.ac.jp

**Abstract:** V–4Cr–xTi (x = 0 to 4) alloys were used to investigate the additional effect of Cr, Ti and interstitial impurities on the microstructural evolution in He-irradiated V–Cr–Ti alloys to minimize radioactivity after fusion neutron irradiation. Transmission electron microscopy and atom probe tomography were carried out to the He-irradiated specimens at 500 °C with 0.5 dpa at peak damage. A flash electro-polishing method for the FIB-extracted specimen was established for the ion-irradiated vanadium alloys. The microstructural evolution of the irradiation-induced titanium-oxycarbonitride, Ti(CON) precipitates was observed and was influenced by the effect of Ti addition on the Ti(CON) precipitation. Apparent Ti(CON) precipitates formed in V-4Cr-xTi with 2% addition of Ti. In the V-4Cr-1Ti alloy, a high density Ti enriched cluster was formed. The origin of the irradiation hardening increase resulted from the size distribution of Ti(CON) precipitation from the dispersed barrier-hardening theory.

**Keywords:** vanadium alloy; focused ion beam; irradiation hardening; transmission electron microscopy; atom probe tomography; Ti(CON) precipitate

## 1. Introduction

V–4Cr–4Ti alloys are a candidate of blanket structural materials in fusion power systems. Critical issues for vanadium alloys for fusion reactor application, such as high temperature mechanical properties, irradiation-induced swelling and oxidation, have been improved by the addition of Cr and Ti to the matrix [1–5]. But the use of V–Cr–Ti alloys in low-temperature (<400 °C) regimes was restricted by a low-temperature embrittlement during neutron irradiation [6]. To avoid this drawback, highly purified V–4Cr–4Ti alloys named as "NIFS-HEAT-1 and -2", have been developed by the National Institute for Fusion Science (NIFS) [7–10]. It is expected that the NIFS-Heat2 alloy will have significantly lower radioactivity after neutron exposures on the first wall of a fusion reactor, because the reduction of radiation waste material and a cooling period provide economic and safety benefits [11].

Nagasaka proposed that the reduction of the cooling period of V–Cr–Ti alloys for a full remote recycle within 10 years can be achieved by reducing the Ti addition without decreasing the high temperature mechanical property due to the addition of Cr from a V–4Cr–4Ti alloy [12]. The cooling time will be reduced by the reduction of the Ti addition but radiation-induced swelling and a loss of mechanical strength, such as through creep strength and irradiation performance, may arise. Therefore, an optimization of the Ti and Cr modification to the V–Cr–Ti alloys is important for the further development of V–Cr–Ti alloys with a high performance under irradiation to balance the irradiation embrittlement at low temperature and swelling behavior [13].

A new set of V–Cr–Ti alloys with a lower Ti and higher Cr content was fabricated based on V–4Cr–4Ti alloys. Nano-indenter tests were carried out for those alloys irradiated with 2MeV He ions at 500 °C and significant irradiation hardening occurred in V-(4-8)Cr-Ti alloy with 1% Ti addition. This suggests that microstructural changes due to 1% addition of Ti are effective for irradiation hardening in V-Cr-Ti alloys [14]. Though microstructural observation is required for He-ion irradiated V-Cr-Ti alloys due to their irradiation hardening behavior, the preparation of TEM specimens of ion-irradiated vanadium alloys was difficult because the sampling that focused on ion beam milling (FIB) for vanadium alloys was not abundant enough to observe the microstructure due to the artifact produced by Ga ion damage.

In this work, a flash electro-polishing method for the FIB-extracted specimen [15] was established for vanadium alloys and allowed us to observe the true microstructure in ion-irradiated vanadium alloys. Microstructural observations by transmission electron microscopy (TEM) and atom probe tomography (APT) measurements were carried out to investigate the microstructures in V–Cr–Ti alloys after He-ion irradiation. The origin of irradiation hardening and the impurity effect for irradiation hardening in the He-ion irradiated V–Cr–Ti alloys was discussed based on the correlation between the irradiation hardening behavior and microstructural changes after He irradiation.

## 2. Materials and Methods

Nine types of V–4Cr–(0–4)Ti ternary alloys and V-8Cr-2Ti alloy were fabricated. The chemical composition of these alloys is shown in Table 1. Two impurity levels of each alloy were prepared: (1) a conventional fabrication containing ~500 ppm of C+N+O interstitial gas impurity, and (2) highly purified V–Cr–Ti alloys, which were marked as "(h)", containing approximately half the interstitial gas impurity in conventional alloys by using highly purified V ingots in fabrication [11]. Thin specimen plates of 10 mm × 2 mm × 0.2 mm were punched out with a die and annealed for 2 h at 1000 °C in a vacuum ($\sim2 \times 10^{-4}$ Pa). The specimens were subjected to 2-MeV $^4$He ion irradiation using a tandem accelerator at the Wakasa Wan Energy Research Center (WERC). Detailed information on the tandem accelerator at the WERC is described elsewhere [14]. The sample stage was heated on a Mo holder with a ceramic heater and maintained within ±5 °C during ion irradiation. Specimens were irradiated at 500 °C with a 0.5 dpa peak at 3.6 μm depth.

**Table 1.** Chemical composition of V–Cr–Ti alloys measured by ICP-AES method. An (h) indicates a highly-purified impurity alloy.

| Composition (wt.%) | Cr | Ti | C | N | O | Mo | Al | Si |
|---|---|---|---|---|---|---|---|---|
| V-4Cr | 3.80 | 0.002 | 0.004 | 0.005 | 0.036 | <0.001 | 0.005 | 0.02 |
| V-4Cr-0.1Ti(h) | 3.90 | 0.09 | 0.007 | 0.003 | 0.017 | <0.001 | 0.011 | 0.02 |
| V-4Cr-1Ti | 3.86 | 0.96 | 0.005 | 0.006 | 0.035 | <0.001 | 0.006 | 0.016 |
| V-4Cr-1Ti(h) | 4.02 | 0.96 | 0.008 | 0.004 | 0.016 | <0.001 | 0.009 | 0.02 |
| V-4Cr-2Ti | 3.94 | 1.93 | 0.005 | 0.005 | 0.037 | <0.001 | 0.005 | 0.02 |
| V-4Cr-2Ti(h) | 3.89 | 1.92 | 0.008 | 0.003 | 0.015 | <0.001 | 0.006 | 0.02 |
| V-4Cr-3Ti | 3.81 | 2.99 | 0.004 | 0.004 | 0.037 | <0.001 | 0.007 | 0.02 |
| V-4Cr-3Ti(h) | 3.92 | 2.99 | 0.009 | 0.003 | 0.016 | <0.001 | 0.007 | 0.02 |
| V-4Cr-4Ti(h) | 4.11 | 3.89 | 0.008 | 0.003 | 0.018 | <0.001 | 0.018 | 0.02 |
| V-8Cr-2Ti(h) | 7.83 | 1.97 | 0.009 | 0.003 | 0.014 | <0.001 | 0.013 | 0.02 |

A damage profile and a He-ion range in vanadium were plotted in Figure 1 as calculated by using the SRIM-2013 code, which calculates the transport and energy loss of ions in matter [16]. The displacement threshold energy was 40 eV.

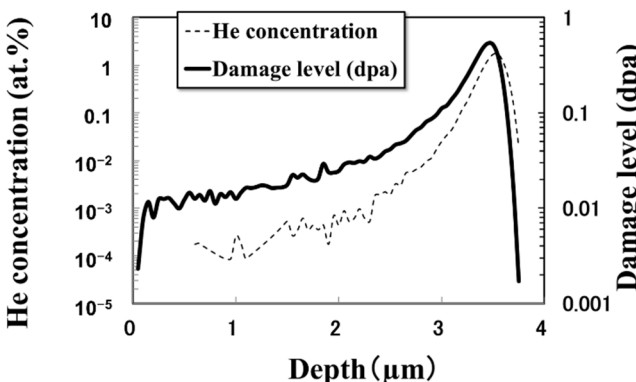

**Figure 1.** Calculation results of an ion range profile of implanted He ion and a vacancy concentration profile in V, obtained by 2-MeV He$^+$-ion irradiation.

A thin plate with a size of $15 \times 10 \times 0.2$ mm was extracted from the surface of the irradiated specimen by a focused ion beam (FIB, JEOL JIB-4500) to fabricate a specimen. A Ga ion with 30 keV was used. When the extracted FIB specimen was observed using TEM, the thin damaged layer caused by Ga-ion irradiation due to FIB fabrication has often hindered the true image of irradiation-induced damage that is produced by high-energy ion-particle irradiation. A method for surface cleaning by an Ar ion beam with 0.5 to 2 keV using a precision ion polishing system, "PIPS" or Gentle Mill was usually used to remove the artifacts from Ga ion beam but it has failed. To remove artifacts from Ga-ion irradiation, we developed a flash electropolishing method to clean the surface of the extracted FIB specimen. A Mo grid mesh was used to load the extracted FIB specimen. The extracted FIB specimen on the Mo grid mesh was electropolished at $-40\,^\circ$C for 0.4 s at 10 V in an electrolyte solution of methanol with sulfuric acid. The distance between the specimen and the electrode was maintained at a constant 20 mm. Although some specimens lost a shallow surface area due to excessive cleaning by electropolishing with an overcurrent, the artifact created by Ga-ion damage was eliminated on the specimen surface. This was confirmed by the clear image of the unirradiated area beyond a 4 μm depth, where an implanted He ion could not reach.

It is currently difficult to observe the true irradiation damage of ion-irradiated vanadium alloys using cross-sectional FIB technology for specimen preparation because of artifacts that are produced by Ga-ion irradiation. Because most of the samples were prepared successfully (nine of the 11 specimens that were fabricated), a flash electro-polishing method for the FIB-extracted specimen was established. Although a shallow area (~0.5 μm) was sometimes lost, the artifacts created by Ga-ion irradiation were removed completely. Successful cleaning of the damaged layer was achieved by a good temperature control at a constant $-40\,^\circ$C with a current control to maintain the distance between the specimen and an electrode. When the timer set was adjusted to an optimum condition in the flash electropolishing method, the fabrication of FIB-extracted specimens with other irradiation conditions became available without specimen loss for the trials.

The samples were observed using a JEOL JEM-2100TM at the University of Fukui at 200 kV and a Hitach HF-3000 in INSS at 300 kV at room temperature. The TEM-FIB specimen thickness was obtained by a sample thickness determination using a convergent bean electron diffraction method (CBED) [17] with a g = (200) or (110) reflection for V–Cr–Ti alloys.

Needle samples were prepared for APT from a 3 μm depth of the irradiated V–4Cr–1Ti at 500 °C with a conventional impurity level by microsampling using a FIB. An APT observation was carried out at the RI facility in the "Fugen" site in JAEA Tsuruga [18]. The APT measurement was carried out by using a LEAP-3000XHR instrument applying a laser power of 0.2 nJ, with a laser pulse repetition rate of 250 kHz, a DC voltage in the range from 2 to 8 kV, and a specimen temperature of 35 K. A few hundred million atoms were typically collected for each sample. Cameca IVAS$^\circledR$ software was used for the APT data

reconstruction. Analytical parameters in the IVAS software were set to d-max = 0.7 nm for the region of interest (ROI) from the core position of a cluster, Nmin = 30 ions for the minimum number of atoms in a cluster, and NNO = 1 for the nearest neighbor order, the number of atoms in a cluster within the d-max.

After He-ion irradiation, a nano-indentation test was conducted at room temperature by using a Elionix ENT-1100a and a Elionix ENT-2100 (Elionix Inc., Japan) nano-indenter with a Berkovich diamond indenter tip and a direction of indentation parallel to the ion beam axis, which is normal to the irradiated surface. The nano-indenter test was carried out with an indentation depth of 500 nm. The indentation depth was determined from the effective depth where the plastic and elastic deformed area expanded during the nano-indenter test and corresponded to five times the length of the nano-indenter depth. The result of nano-indentation test was published in detail in our previous work [14].

## 3. Results

### 3.1. TEM Observation

Figure 2 shows cross-sectional TEM images of He-irradiated V–Cr–Ti(h) alloys. Tangled dislocations and a high density of He bubbles formed at a ~3.5–3.8 μm depth in all alloys. In the V–4Cr–0.1Ti(h) alloy, no defect cluster formed at a 1 μm depth and large dislocation loops with He bubbles at the center of loops formed at a 3 μm depth. Large dislocation loops lay on the {111} habit plane from the electron diffraction analysis. A high density of tiny defect clusters of a 2–3 nm diameter formed homogenously from the surface to the peak damage area in the V–4Cr–1Ti(h) alloys distinct from the V–4Cr–0.1Ti(h) alloy. Large dislocation loops on the {111} plane formed in the mid-range from 1 to 3 μm but the density of the large loops of ~100 nm diameter was lower than that of the tiny defect clusters. The tiny defect clusters could not be characterized because they were small and minimal information was obtained by electron diffraction analysis. Detailed information on the characterization of tiny clusters was explained by using APT data.

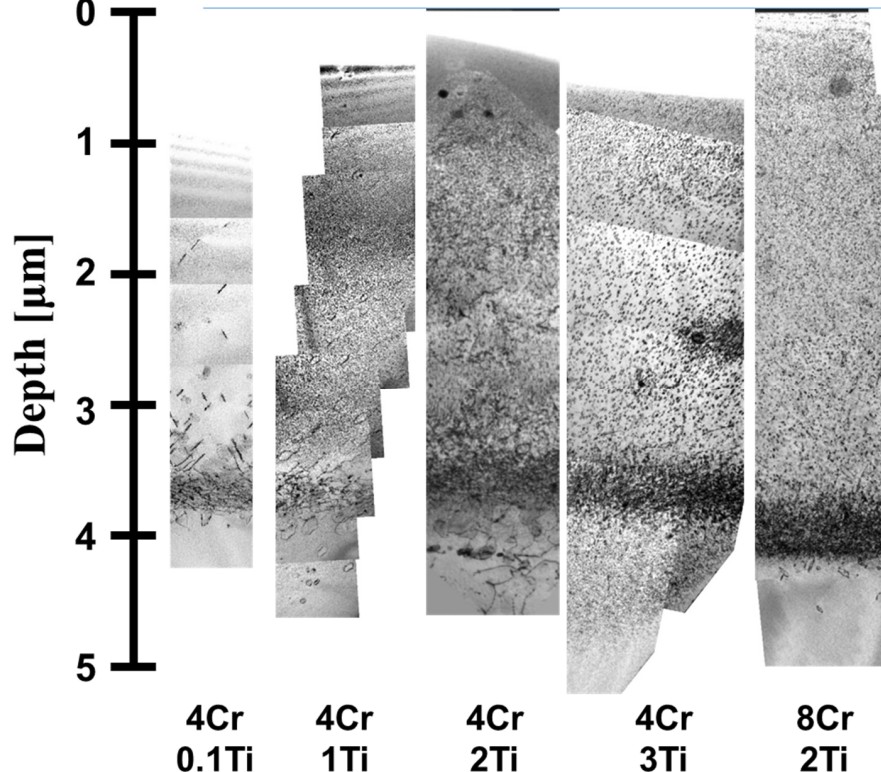

**Figure 2.** Cross-sectional bright-field TEM images of He-irradiated V–Cr–Ti(h) alloys at 500 °C from the irradiation surface to the peak damage area (~3.8 μm).

In the V–4Cr–2Ti(h) and V–4Cr–3Ti(h) alloys, the high-density platelets lay on the {100} plane from the surface to the peak damage region. This platelet was a type of titanium-oxycarbonytride, Ti(OCN) precipitate from electron diffraction analysis. Ti(CON) precipitate has a cubic TiO rock-salt structure (a = 0.417 nm) in which the sub-lattice site of anion atom is occupied by the oxygen, carbon, and nitrogen. It showed a 3/4[200] diffraction spot of Ti(CON) precipitate on the [001] zone axis projection of vanadium matrix [19–24]. At a 2.5 µm V–4Cr–3Ti(h) alloy depth, a bulky Ti(CON) precipitate was dissolved and new tiny Ti(CON) platelets were nucleated and formed around the bulky Ti(CON) precipitate. Radiation-induced segregation phenomena occurred during He-ion irradiation. For the V–8Cr–2Ti(h) alloy, the same microstructural feature appeared with V–4Cr–2Ti(h) and V–4Cr–3Ti(h) alloys. Cr addition in the V–Cr–Ti alloys did not affect the microstructural morphology for the formation of radiation-induced defect clusters at a 500 °C irradiation. This damage profile affected the nano-indentation data quality. As mentioned in the experimental procedure section, the plastic and elastic deformed area by nano-indentation was expanded to a depth that corresponds to five times the length of the nano-indenter depth, which was 3 µm in this work. Therefore, the indentation hardness data does not involve irradiation-hardening data at the peak damage area where tangled dislocations and many tiny He bubbles formed during He irradiation.

The detailed microstructural images at a shallow area (1 µm depth) and close to the peak damage area (3 µm depth) in the V–Cr–Ti(h) alloys with a highly purified impurity level are shown in Figure 3. The estimated damage level at 1 µm and 3 µm was 0.02 dpa and 0.1 dpa, respectively, as shown in Figure 1. The density of the tiny defect clusters in the V–4Cr–1Ti(h) and Ti(CON) precipitate in the V–4Cr–2Ti(h) and V–4Cr–3Ti(h) at a 1 and 3 µm depth did not change much, even though the damage level at a 3 µm depth was five times larger than that at a 1 µm depth. Figure 4 shows the microstructural images at a 1 and 3 µm depth in the V–Cr–Ti alloys with conventional impurity levels.

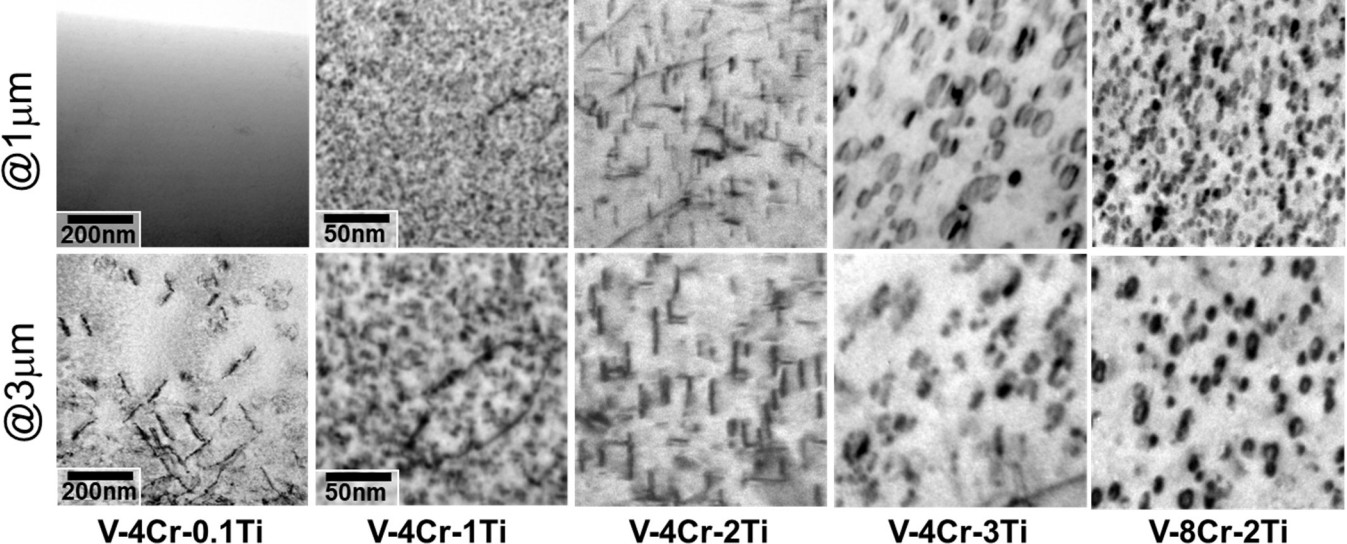

**Figure 3.** Bright-field TEM images of He-irradiated V–Cr–Ti(h) alloys at 500 °C. Upper parts show high-magnification images at a 1 µm depth and lower parts show images at a 3 m depth. Note that the left photo has a different magnification than the other ones.

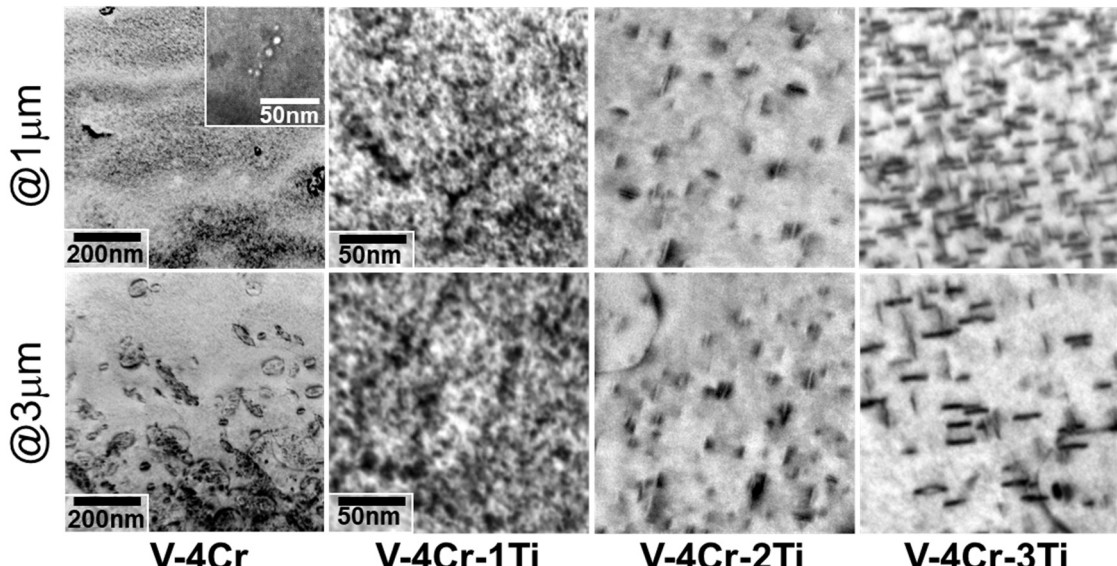

**Figure 4.** Bright-field TEM images of He-irradiated V–Cr–Ti alloys at 500 °C. Upper parts show high-magnification images at 1-m depth and lower parts show images at a 3 µm depth. An inserted image of the V–4Cr at a 1 µm depth shows a void image. Note that the left photo (V-4Cr) has a different magnification than the other ones.

The tendency for microstructural evolution after He-ion irradiation in V–Cr–Ti alloys with a conventional impurity level was almost the same as that in the V–Cr–Ti(h) alloys with a highly purified impurity level.

In the V–4Cr alloy, cavities such as voids or He bubbles were formed at a 1 µm depth and {111} dislocation loops with cavities at the center of the loop formed at a 3 µm depth. The density of the tiny defect clusters formed from the surface area to the peak damage area in the V–4Cr–1Ti. Ti(CON) precipitates on the {100} habit plane were formed from the surface area to the peak damage area in the V–4Cr–2Ti and V–4Cr–3Ti alloys. The density and size distribution of the defect clusters and Ti(CON) precipitate were measured from the photographs in Figures 3 and 4. Figure 5 shows the density and mean size of the tiny defect clusters and Ti(CON) precipitates that formed at a 2.5 to 3 µm depth in the V–Cr–Ti(h) with a highly purified impurity level and V–Cr–Ti alloys with a conventional impurity level. The difference in mean size of the Ti(CON) precipitates was different between the alloys with a highly purified impurity level and those with a conventional impurity level, and the mean precipitate size in the V–4Cr–2Ti and –3Ti alloys with a conventional impurity level was larger than that with a highly purified impurity level.

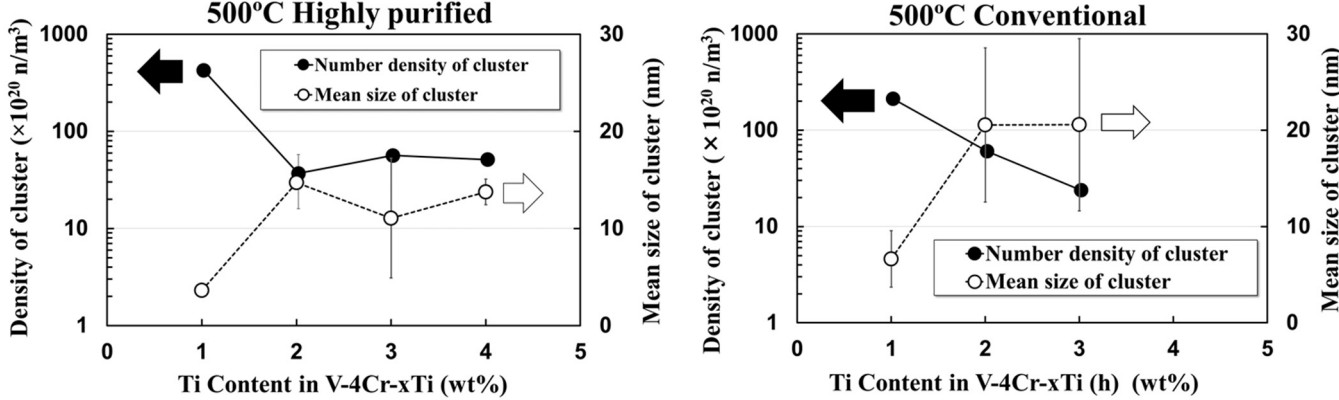

**Figure 5.** Number density and mean size of defect clusters in V–Cr–Ti alloys irradiated at 500 °C as a function of Ti content in V–4Cr–xTi. Left plot for V–Cr–Ti(h) alloys with highly purified impurity level and right plot for V–Cr–Ti alloys with a conventional impurity level.

### 3.2. APT Observation

An APT measurement was performed to characterize the tiny defect clusters in the V–4Cr–1Ti alloy with a conventional impurity level after He-ion irradiation. The left part of Figure 6 shows the APT measurement results and a sliced image of the three-dimensional elemental map of the Cr, Ti, C and O in the V–4Cr–1Ti alloy that was irradiated at 500 °C. Elemental Cr and O were distributed homogeneously in a matrix. Elemental Ti formed an aggregated-like platelet with an average of 3 nm and the Ti aggregate density was approximately $2 \times 10^{24}$ /m$^3$. The tiny defect cluster density that was visible in the TEM observation was obtained as $4 \times 10^{22}$ /m$^3$. From the chemical component analysis of local area in APT measurement, the ratio of atom concentration inside the Ti aggregate was obtained. The right of Figure 6 shows the ratio of atom concentration inside 15 Ti aggregates that were observed in the He-ion irradiated V–4Cr–1Ti with a conventional impurity level using APT measurements.

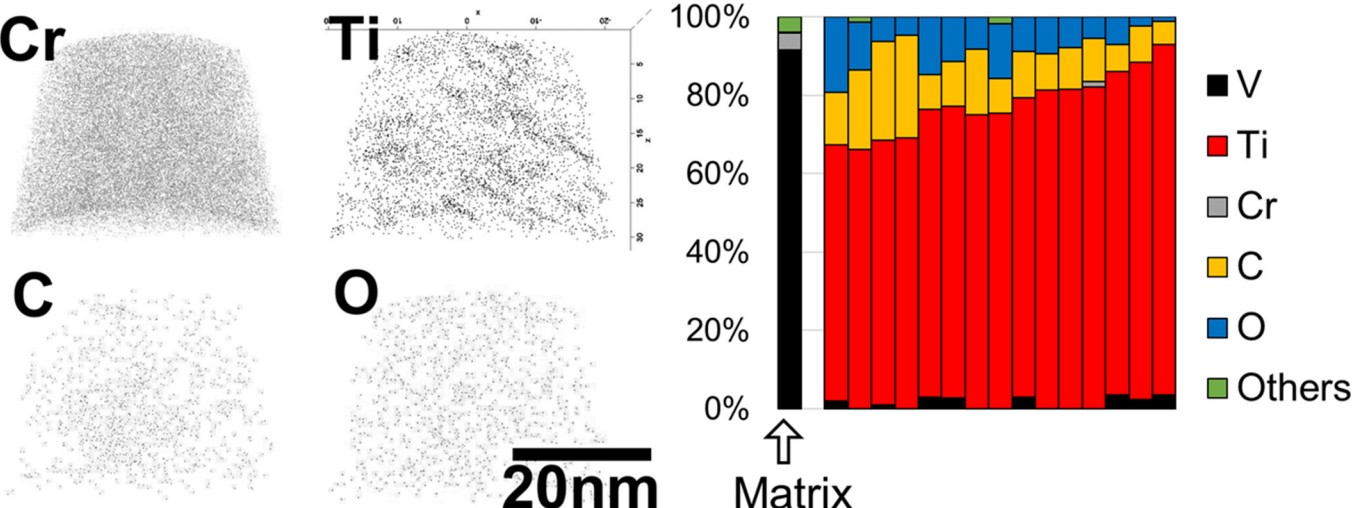

**Figure 6.** APT measurement in He-irradiated V–4Cr–1Ti alloy at 500 °C with a conventional impurity level. The left section shows atom maps in an irradiated sample. The right section shows the ratio of atom concentrations inside each Ti aggregate by APT measurement. Data on the left end shows the chemical composition in the matrix.

A data at the left end in a histogram of Figure 6 shows the component ratio of alloy matrix, and the others show the component ratio inside Ti aggregates. The main elemental Ti aggregates were composed of Ti with a few interstitial impurities of C and O. The average ratio of each element in the Ti aggregate was obtained as V:Ti:Cr:O:C = 1:77:0:13:9. Metallic elements were only occupied by elemental Ti in the neutron-irradiated V–4Cr–4Ti alloys at 350 °C and 450 °C [25,26], which are in agreement with the result of this work. The Ti distribution in the Ti aggregates was likely to be independent of the oxygen distribution in the specimen, but the Ti aggregate contained a certain amount of interstitial impurities, such as oxygen and carbon.

### 3.3. Nano-Indentation Test

A significant irradiation hardening occurred in V-(4–8)Cr-xTi with a 1% Ti addition that was irradiated at 500 °C. The detailed data on irradiation hardness increases has been published elsewhere [14]. Here, the increase in irradiation hardness results for the He-irradiated V–Cr–Ti alloys at 500 °C was replotted in Figure 7 with more data that was newly obtained. The left and right figures show the nano-indentation hardness of the unirradiated V–Cr–Ti alloys with conventional and highly purified impurity levels, respectively. Figure 7 shows that the irradiation hardness was changed significantly by the Ti content in the V–Cr–Ti alloys. When the Ti addition in the V–(4–8)Cr–Ti alloys increased from 0% to 1%, a larger increase in irradiation hardening was observed. At a 1% addition of

Ti in the V–(4–8)Cr–xTi alloys, the largest irradiation hardening was observed. From 2% to 4% Ti additions in the V–(4–8)Cr–xTi alloys, the irradiation hardening gradually decreased and did not change as much as after an increase in the Ti addition. Therefore 1% Ti addition is the most effective level to produce irradiation hardening in V–(4–8)Cr–Ti alloys. The irradiation hardening increases of the V–(4–8)Cr–1Ti with a conventional impurity level was larger than those of V–(4–8)Cr–1Ti(h) alloys with a highly purified impurity.

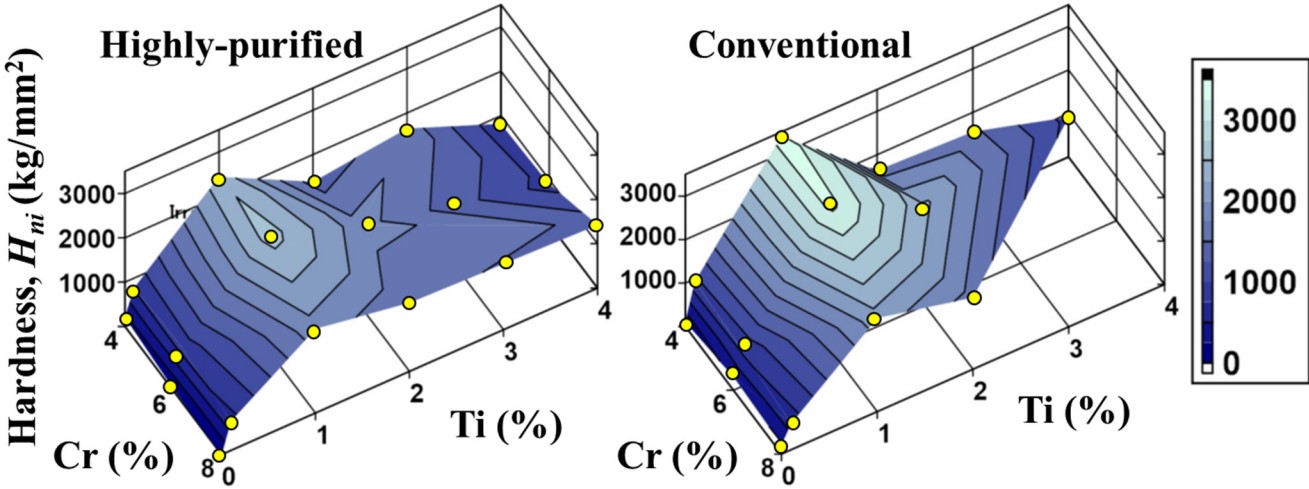

**Figure 7.** Profiles of hardness increases for He-irradiated V–Cr–Ti alloys at 500 °C. The left profile indicates V–Cr–Ti(h) alloys with a highly purified impurity level and the right profile indicates V–Cr–Ti alloys with a conventional impurity level.

## 4. Discussion

To understand the correlation between microstructural change and mechanical property changes in the irradiated materials, the classic dispersed-barrier-hardening (DBH) model was used to evaluate the irradiation hardening that was connected to the microstructural information by applying the density and size of defect variants [27,28]. For a correct statistical estimation of the defect cluster density from the TEM images, the measured density was weighted by using the visibility of defects in a TEM image [21,29]. The tangled dislocation and dislocation loops were negligible because the dislocation density in the midrange of the ion damage profile was low and only Ti(CON) precipitate was available for estimating irradiation hardening. Because one-third of the Ti(CON) should be invisible by the g = [110] reflection in the bright-field image under TEM observation, the Ti(CON) density should be weighted by a factor of 1.5. To obtain the irradiation hardening value from the microstructural changes in the material, a dispersed barrier model was used, and the yield stress increased because of the microstructural changes that are expressed in Equation (1):

$$\Delta\sigma_y = \sqrt{\sum \left( M\alpha_i \mu b \sqrt{N_i D_i} \right)^2} \tag{1}$$

where M is the Taylor factor (3.06 for polycrystals); $\mu$ is the shear modulus of the matrix (46 GPa); b is the Burgers vector (0.261 nm); $\Delta\sigma_y$ is the increment in yield strength due to $N_i$ number density, $D_i$ size of discrete defects, and $a_i$ barrier strength of discrete defects of type i such as Ti(CON) precipitates and dislocation loops. The data from the Ti aggregates observed by APT measurements were not used and only the TEM data for the TEM-visible defect clusters were used in this estimation. When *C* is a conversion parameter, its nano-indentation hardness to yield stress can be used to determine the equivalent relationship as $\Delta\sigma_{y-total}^{Calc} \approx C \cdot \Delta H_{it}$, through which a value of *C* was confirmed by $\Delta H_{it}/\Delta\sigma_{y-total}^{Calc}$. The value of *C* was determined as 0.09 from other work on self-ion-irradiated stainless steel [30]. The values of M, $\mu$, and b are fixed parameters that are specific to a material, and the values of *N* and *D* are provided by the measurement of microstructural changes in the

irradiated specimens. Previous studies have reported that the value of $\alpha$ for the defect cluster ranges from 0.34 to 0.56 [20,21,29,31], and the barrier strength $\alpha$ for the Ti(CON) precipitate was 0.4 in this work after fitting the data with irradiation hardness changes. The barrier strength for the dislocation loops was 0.35 in this work. For this calculation, it was not considered that He bubbles and tangled dislocations at the peak position of the He ion range did not contribute to irradiation hardening in the nano-indentation test because it is assumed that the depth area affected by the deformation due to the nano-indentation did not reach the peak position. It was apparent from the experimental results that the He-irradiated V-4Cr and V-4Cr-0.1(h) alloys showed low irradiation hardening, regardless of the existence of highly damaged area with He bubbles and tangled dislocations near the ion range at 3.5~3.8 μm. As a result, the contribution of damage structures at the peak position of the He ion range was ignored in this calculation of irradiation hardening estimation by using the DBH model.

Figure 8 presents a plot of the balance of the increase in yield stress in the V–Cr–Ti(h) and V–Cr–Ti alloy after ion irradiation. The measured irradiation hardening data and the estimated yield stress increases are shown by open circles and a bar chart, respectively. A gray and black bar chart shows hardening contribution of Ti(CON) precipitates and dislocation loops, respectively. Most of irradiation hardening was contributed to by the Ti(CON) precipitate, as shown in Figure 8. The correlation between the measured and estimated yield stress increased in the V–Cr–Ti(h) alloys with a highly purified impurity level, thereby showing good agreement. Additionally, Ti(CON) precipitates were found to be important for the irradiation hardening behavior in the V–Cr–Ti(h) alloys. The measured irradiation hardening was significantly larger than the estimated yield stress increase in the V–4Cr–1Ti alloy with a conventional impurity level, but the V–4Cr–2Ti and 3Ti alloys showed a good agreement between the measured and estimated yield stress increase. The difference in yield stress increase between the measured and estimated yield stress increase is caused by Ti aggregates combining with oxygen impurities in the V–4Cr–1Ti alloy.

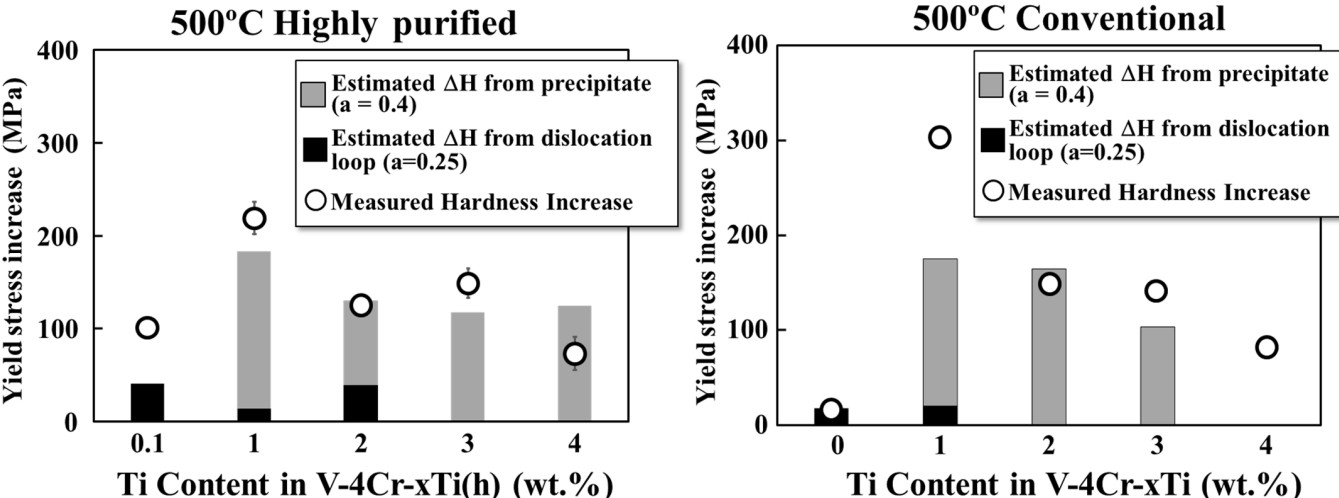

**Figure 8.** Balance of increase in estimated yield stress in V–Cr–Ti alloy after He-ion irradiation. Open symbols show the yield stress increase estimated through measured nano-indentation hardness. Bar charts show the yield stress increase estimated through the dispersed barrier-hardening model from microstructural information obtained from TEM observation. A gray and black bar chart shows the hardening contribution of Ti(CON) precipitates and dislocation loops, respectively.

When it is assumed that the APT data of the Ti aggregates in the V–4Cr–1Ti with a conventional impurity level were used for another defect cluster and contributed to irradiation hardening, a Ti aggregate may have an $\alpha$ value of 0.008 as a weak, resistant obstacle against mobile dislocation. Therefore, the oxygen impurity will combine with a Ti aggregate and strengthen the aggregate from the dispersed-barrier-hardening model

analysis. It has been reported that in the formation of Ti(CON), the easy uptake of oxygen impurities by Ti aggregates is somewhat influenced by the initial impurity level in the base and weld metals of the V–Cr–Ti alloys [29,32]. In conclusion, the Ti(CON) precipitate that is visible in the TEM observation is likely to take up oxygen impurities in the matrix and affect irradiation hardening in the V–Cr–Ti alloys that contain more than 2% Ti.

## 5. Conclusions

Nine V–4Cr–(0–4)Ti alloys and V-8Cr-2Ti alloys were used to investigate the microstructural evolution in V–Cr–Ti alloys during He-ion irradiation. These alloys were subjected to TEM and APT observations after 2-MeV He-ion irradiation at 500 °C with a 0.5-dpa peak damage.

To obtain true microstructural information through TEM observations by using the FIB-extracted specimen, flash electro-polishing for the FIB extracted specimen was almost established in this work. From the cross-sectional TEM observation, tiny defect clusters, such as Ti aggregates and Ti(CON) precipitates, formed from the surface to a 3 μm depth and tangled dislocations and a high density of He bubbles formed at a ~3.5–3.8 μm depth in all alloys. A high density of tiny defect clusters with a 2–3-nm diameter formed homogenously from the surface to the peak damage area in the V–4Cr–1Ti(h) alloys, which is distinct from the V–4Cr–0.1Ti(h) alloy. In the V–4Cr–2Ti(h) and V-4Cr-3Ti(h) alloys, the high density of Ti(CON) platelets that lay on the {100} plane formed homogeneously from the surface to the peak damage region. The tendency of microstructural evolution after He-ion irradiation in the V–Cr–Ti alloys with a conventional impurity level was almost the same as that in the V–Cr–Ti(h) alloys with a highly purified impurity level. An APT measurement was performed to characterize the tiny defect clusters in the V–4Cr–1Ti alloy with a conventional impurity level after He-ion irradiation and it was determined to be a type of Ti aggregate with a certain level of interstitial impurities. This result suggests that a remarkable population of TEM-invisible Ti-rich clusters exist and the oxygen impurities do not aggregate to form Ti(CON) precipitates in the V–4Cr–1Ti alloy with a conventional impurity level.

The results from the microstructural evolution of irradiation-induced Ti(CON) precipitates were influenced by the effect of Ti addition on the Ti(CON) precipitation process. From the dispersed-barrier hardening model, the barrier factor of Ti(CON) was obtained as 0.4. The correlation between the measured and estimated yield stress increase in the V–Cr–Ti(h) alloys with a highly purified impurity level agreed well and the Ti(CON) precipitate affected the irradiation hardening behavior in V–Cr–Ti alloys with a highly purified and conventional impurity level. The Ti(CON) precipitate that was visible in the TEM observation is likely to take up oxygen impurities in the matrix and result in irradiation hardening in V–Cr–Ti alloys that contained more than 2% Ti.

**Author Contributions:** Conceptualization, K.-i.F. and T.N.; TEM observation, K.-i.F.; Sample preparation, T.N., S.M. and Y.K.; Ion irradiation, R.I. and S.M.; Nano indentation, S.M. and Y.K.; Manuscript writing, K.-i.F. All authors have read and agreed to the published version of the manuscript.

**Funding:** This study was partly supported by the NIFS Budget, Code NIFS17KEMF098. This work was supported by a JSPS Grant-in-Aid for Scientific Research (A) 20H00144.

**Acknowledgments:** The authors are grateful to the technical staffs of the tandem accelerator at the Wakasa-Wan Energy Research center for supplying high-quality ion beams. The authors are grateful to K. Fujii at INSS for supporting the TEM and APT observation experiments.

**Conflicts of Interest:** The authors declare no conflict of interest.

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
