# Peer review of "Correlation between Microstructural Change and Irradiation Hardening Behavior of He-Irradiated V–Cr–Ti Alloys with Low Ti Addition"

_qubs, doi:10.3390/qubs5030026_

Round 1
Reviewer 1 Report
The manuscript qubs-1323679 entitled „Correlation between microstructural change and irradiation hardening behavior of He-irradiated V–Cr–Ti alloys with low Ti addition” deals with the effects of He ion irradiation in vanadium-based alloys. The nanometre scaled microstructural changes / defects were investigated with advanced methods (TEM and APT). Furthermore, nanoindentation results are reported to compare the irradiation hardening depending on the composition and purity of the different alloys.
General remarks:
The structure of the manuscript contains a result and discussion part. However, there are results presented in the discussion part for the first time (Fig. 7 and Fig. 8.). It would be better to show the data in the results section.
The quality of the images should be improved. E.g. the scale on the left side of Fig.2 is too large. Also the labels (4Cr 0.1Ti…) are too large. The same applies for other Figures.
The figures have to be named in the text before the figures are shown. This isn´t case in the document and must be corrected.
A Schematic representation will be helpful to classify the microstructural changes / defects (Ti(CON), Ti aggregates, He bubbles…) related to the alloy composition and impurity level.
The exact nature of the so called Ti(CON) precipitates remains unclear. E.g. what crystal structure?
Line 201: The Figure 6 is not clear and what does the lower part exactly show? Why is no space resolved information provided such as proximity histogram. APT can provide valuable information but the quality of the APT data evaluation and presentation should be improved.
Minor revisions:
There are a lot of minor mistakes that have to be corrected, e.g.:
Line 10: “interstitial impurities” instead of “gas impurities”?
Line 13: delete “observation”
Line 21: do not use the abbreviations (FIB, TEM, APT) here and delete "sampling" and "observation"
Line 26: delete the repeated “alloys”
Line 31: what is the meaning of “a.k.a, NIFS-HEAT-1 and -2”
Line 58: write “carried out”
Line 65: write “ is shown”
Line 76: explain what is “SRIM-code”
Line 78: which methods were used to determine the composition in table 1
Line 97 and in many other cases: here is a wrong symbol for µ
Line 114: The information about the preparation of the FIB preparation is redundant several times (also line 242) see where is can be deleted best
Line 116: don´t write “81%” but something like “most of”
Line 129: The figure has to be named in the text before the figure is shown. This is the case several times in the document and should be corrected.
Line 166 (Figure 5): upper diagram: “Density of custre…” instead of “Density of cluster
….”
Line 229: “…have been reported…”. Is there a reference missing?
Author Response
Reviewer 1
The manuscript qubs-1323679 entitled „Correlation between microstructural change and irradiation hardening behavior of He-irradiated V–Cr–Ti alloys with low Ti addition” deals with the effects of He ion irradiation in vanadium-based alloys. The nanometre scaled microstructural changes / defects were investigated with advanced methods (TEM and APT). Furthermore, nanoindentation results are reported to compare the irradiation hardening depending on the composition and purity of the different alloys.
General remarks:
- The structure of the manuscript contains a result and discussion part. However, there are results presented in the discussion part for the first time (Fig. 7 and Fig. 8.). It would be better to show the data in the results section.
Response 1: Thank you for your advice. I move the hardness data to the results section.
- The quality of the images should be improved. E.g. the scale on the left side of Fig.2 is too large. Also the labels (4Cr 0.1Ti…) are too large. The same applies for other Figures.
Response 2: I revised the size of labels in figures according to the reviewer’s comment.
- The figures have to be named in the text before the figures are shown. This isn´t case in the document and must be corrected.
Response 3: I revised and adjusted the position of figures after the text where describes the figures.
- A Schematic representation will be helpful to classify the microstructural changes / defects (Ti(CON), Ti aggregates, He bubbles…) related to the alloy composition and impurity level.
Response 4: Thank you for the advice but there is no space and time to create the figure. I understand it will be helpful to understand it all easily….
- The exact nature of the so called Ti(CON) precipitates remains unclear. E.g. what crystal structure?
Response 5: I revised the explanation of Ti(CON) precipitate in details as follows; “This platelet was a type of Ti(OCN) precipitate from electron diffraction analysis. Ti(CON) precipitate has a cubic TiO with a rock-salt structure (a=0.417nm) and the sub-lattice site of anion atom is occupied by the oxygen, carbon and nitrogen. It showed a 3/4[200] diffraction spot of Ti(CON) precipitate on the [001] zone axis projection of vanadium ma-trix [17-22].”
- Line 201: The Figure 6 is not clear and what does the lower part exactly show? Why is no space resolved information provided such as proximity histogram. APT can provide valuable information but the quality of the APT data evaluation and presentation should be improved.
Response 6: I added the complemental texts that describe the details for Figure 6 as follows;
“Elemental Ti formed an aggregated-like platelet of an average 3 nm and the Ti aggregate density was approximately 2 × 1024 /m3. The tiny defect cluster density that was visible in the TEM observation was obtained as 4 × 1022 /m3. From the chemical component analysis of local area in APT measurement, the ratio of atom concentration inside the Ti aggregate was obtained. The bottom of Figure 6 shows the ratio of atom concentration inside Ti ag-gregate that was observed in the He-ion irradiated V–4Cr–1Ti with a conventional impu-rity level using APT measurements. A data at the left end in Figure 6 shows the compo-nent ratio of alloy matrix, and the others show the component ration inside Ti aggregates. The main elemental Ti aggregates were composed of Ti with a few interstitial impurities of C and O. The average ratio of each element in the Ti aggregate was obtained as V:Ti:Cr:O:C = 1:77:0:13:9.”
- Minor revisions:
There are a lot of minor mistakes that have to be corrected, e.g.:
Line 10: “interstitial impurities” instead of “gas impurities”?
Response 7-1: I revised it according to the reviewer’s comment.
Line 13: delete “observation”
Response 7-2: I revised it according to the reviewer’s comment.
Line 21: do not use the abbreviations (FIB, TEM, APT) here and delete "sampling" and "observation"
Response 7-3: I revised it according to the reviewer’s comment.
Line 26: delete the repeated “alloys”
Response 7-4: I revised it according to the reviewer’s comment.
Line 31: what is the meaning of “a.k.a, NIFS-HEAT-1 and -2”
Response 7-5: I revised it as follows; named as “NIFS-Heat-1 and -2”
Line 58: write “carried out”
Response 7-6: I revised it according to the reviewer’s comment
Line 65: write “ is shown”
Response 7-7: I revised it according to the reviewer’s comment
Line 76: explain what is “SRIM-code”
Response 7-8: I added the explanation of SRIM code as follows; “SRIM-2013 code which calculate the transport and energy loss of ions in matter [12].”
Line 78: which methods were used to determine the composition in table 1
Response 7-9: I added the method in the table caption as follows; “Chemical composition of V–Cr–Ti alloys measured by ICP-AES method.”
Line 97 and in many other cases: here is a wrong symbol for µ
Response 7-10: Thank you for the check. I revised them all.
Line 114: The information about the preparation of the FIB preparation is redundant several times (also line 242) see where is can be deleted best
Response 7-11: I deleted the first paragraph of discussion section which is mentioned about the superiority of flash electropolishing to remove the duplication about FIB sample preparation, I combined the FIB explanation in the materials and method part with one in the result part to reduce the redundancy.
Line 116: don´t write “81%” but something like “most of”
Response 7-12: I revised it according to the reviewer’s comment
Line 129: The figure has to be named in the text before the figure is shown. This is the case several times in the document and should be corrected.
Response 7-13: I revised and adjusted the position of figures after the text where describes the figures.
Line 166 (Figure 5): upper diagram: “Density of custre…” instead of “Density of cluster….”
Response 7-14: I revised it according to the reviewer’s comment
Line 229: “…have been reported…”. Is there a reference missing?
Response 7-15: I revised the references of the previous works in the same paragraph at all and relabeled the reference numbers.
Reviewer 2 Report
This is a very interesting paper concerning a blanket material for fusion reactors. The experiment set up is very thorough and well thought out and the results are very interesting. However, there are many issues with the results and discussions throughout the manuscript and major revisions are required before it is at an acceptable level for publications.
- Materials & Methods
- Page 3: in Figure 1, helium concentration axis needs unit labelling. Author needs to specify if its in appm or at.%
- The authors need to to specify which model they used in SRIM and where they sourced the displacement energies for the damage calculations.
- The authors need to provide more detail on the process by which they fabricate TEM samples. What Ga ion energies do they use throughout the fabrication process and the size of the FIB’d sample. Ga ion damage can be minimised by use of low energy cleaning at 2kV or 5kV. I am not sure whether the authors have considered this as it is widely used by scientists studying radiation damage in materials to minimise introduction of gallium ion damage.
- The section on atom prob tomography does not provide any detail on the stage temperature during atom probe analysis. Also the authors need to specify whether its in laser or voltage mode and the oscillation frequency.
- In identifying clusters in atom probe tomography, one needs to do a full calibration by testing different Dmin and Nmin values. This is because use of arbitrary Dmin and Nmin values lead to results with low degree of confidence. A full calibration needs to be provided in the main body of the paper or appendix to justify the use of the Dmin and Nmin values otherwise the data presented would have low confidence level. Please consider the following book which is widely recognised by atom probe microscopists.
https://www.springer.com/gp/book/9781461434351
- Results
- The authors need to go through this section and properly label the depth as a lot of the times it comes out as “”.
- The authors need to keep this section strictly to reporting results and move speculative statements to discussion. Some examples of speculative statements are as follows:
- (page 6) “It is likely that the point-defect concentration may be homogenized and the nucleation and growth of defect clusters was flattened from the damage peak area to a shallow surface area by a large bias of point defect flux.”
- (page 7) “It is likely that the 197 Ti(CON) precipitate size was related to the impurity level because the Ti(CON) precipitate consists of a sub-lattice of oxygen and other impurities and more impurities may contribute to Ti(CON) precipitate growth.”
- (page 8) “The results suggest that a remarkable population of TEM-invisible Ti-rich clusters is present and the oxygen impurities do not aggregate to form Ti(CON) precipitates in the 225 V–4Cr–1Ti alloy with a conventional impurity level.”
- The following section on page 5 is confusing: “This damage profile affected the nano-indentation data quality. As mentioned in the experimental procedure section, the plastic and elastic deformed area by nano-indentation was expanded to a depth that corresponds to five times the length of the nano-indenter depth, which was 3 m in this work. Therefore, the indentation hardness data does not involve irradiation-hardening data at the peak damage area where tangled dislocations and many tiny He bubbles formed during He irradiation.” There was no mention of nanoindentation in the methodology section but now the authors are referencing nanoindentation. Also there is no presentation of nanoindentation results. If the authors indent to 300nm depth the indent volume could encompass the irradiation region and they may compare the hardness obtained at that depth with the corresponding unirradiated sample. The difference in hardness between the two values by-passes the size effect experienced at 300nm.
- Do the authors intend on characterising the helium bubbles introduced or are they purely interested in the tail of the irradiation profile where it is mostly displacement damage?
- Discussion
- The authors need to be more precise in comparing neutron irradiation experiments with the helium irradiation performed. For example, what neutron damage was introduced what alloys were studied. What was the size and number density of the various defects and precipitates and how did it compare with the values measured by helium ion irradiation.
- In applying the dispersion hardening model to estimate the increase in yield strength the authors ignores the loop density as well as helium bubble density and only uses the precipitate density which is in accurate. The entire section needs to be re-written including contributions from helium and loops.
Author Response
Reviewer 2
This is a very interesting paper concerning a blanket material for fusion reactors. The experiment set up is very thorough and well thought out and the results are very interesting. However, there are many issues with the results and discussions throughout the manuscript and major revisions are required before it is at an acceptable level for publications.
- Materials & Methods
- Page 3: in Figure 1, helium concentration axis needs unit labelling. Author needs to specify if its in appm or at.%
Response 1: I revised the label in Figure1 as “He concentration (at.%)”.
- The authors need to to specify which model they used in SRIM and where they sourced the displacement energies for the damage calculations.
Response 2: I revised the text concerned with SRIM code calculation as “A damage profile and a He-ion range in vanadium were plotted in Figure 1 as calculated by using SRIM-2013 code which calculate the transport and energy loss of ions in matter [12]. The displacement threshold energy was used as 40 eV.”.
- The authors need to provide more detail on the process by which they fabricate TEM samples. What Ga ion energies do they use throughout the fabrication process and the size of the FIB’d sample. Ga ion damage can be minimised by use of low energy cleaning at 2kV or 5kV. I am not sure whether the authors have considered this as it is widely used by scientists studying radiation damage in materials to minimise introduction of gallium ion damage.
Response 3: I revised and added the explanation of FIB specimen preparation as follows; “A thin plate with a size of 15 x 10 x 0.2 m was extracted from the surface of the irradiated specimen by a focused ion beam (FIB, JEOL JIB-4500) to fabricate a specimen. A Ga ion with 30 keV was used. When the extracted FIB specimen was observed using TEM, the thin damaged layer by Ga-ion irradiation because FIB fabrication has often hindered the true image of irradiation-induced damage that is produced by high-energy ion-particle irradiation. A method for surface cleaning by Ar ion beam with 0.5 to 2 keV using PIPS or Gentle Mill was used to remove the artifacts from Ga ion beam but it has been failed.”
- The section on atom prob tomography does not provide any detail on the stage temperature during atom probe analysis. Also the authors need to specify whether its in laser or voltage mode and the oscillation frequency.
Response 4: I revised and added the explanation of ATP measurement as follows; “The APT measurement was carried out by using a LEAP-3000XHR instrument applying a laser power of 0.2 nJ, with a laser pulse repetition rate of 250 kHz, a DC voltage in the range from 2 to 8 kV, and a specimen temperature of 35 K.”
- In identifying clusters in atom probe tomography, one needs to do a full calibration by testing different Dmin and Nmin values. This is because use of arbitrary Dmin and Nmin values lead to results with low degree of confidence. A full calibration needs to be provided in the main body of the paper or appendix to justify the use of the Dmin and Nmin values otherwise the data presented would have low confidence level. Please consider the following book which is widely recognised by atom probe microscopists.
https://www.springer.com/gp/book/9781461434351
Response 5: I guess that Dmin is Dmax in the reviewer’s comment. We know the importance of Dmax and Nmin and I agree the point that the reviewer suggested. The data of APT measurement was obtained by only one sample so that it is difficult to do a full calibration for vanadium alloys, The parameter determination of Dmax, Nmin, Order, L and derosion was referred to the results of stainless steels basically. In paticular, Dmax was selected from 0.5 to 0.8 for imaging the clusters and determined as 0.7. Here all I want to say in APT measurement section is the fact that Ti aggregates invisible in TEM observation are formed and confirmed by APT measurement qualitatively. Then I just describe the parameter of dmax, Nmin and Order in the text.
- Results
- The authors need to go through this section and properly label the depth as a lot of the times it comes out as “”.
Response 6: Thank you for the check. I revised them all.
- The authors need to keep this section strictly to reporting results and move speculative statements to discussion. Some examples of speculative statements are as follows:
- (page 6) “It is likely that the point-defect concentration may be homogenized and the nucleation and growth of defect clusters was flattened from the damage peak area to a shallow surface area by a large bias of point defect flux.”
Response 7: I delete this text in the result section..
- (page 7) “It is likely that the 197 Ti(CON) precipitate size was related to the impurity level because the Ti(CON) precipitate consists of a sub-lattice of oxygen and other impurities and more impurities may contribute to Ti(CON) precipitate growth.”
Response 8: I delete this text in the result section.
- (page 8) “The results suggest that a remarkable population of TEM-invisible Ti-rich clusters is present and the oxygen impurities do not aggregate to form Ti(CON) precipitates in the 225 V–4Cr–1Ti alloy with a conventional impurity level.”
Response 9: I delete this text in the result section.
- The following section on page 5 is confusing: “This damage profile affected the nano-indentation data quality. As mentioned in the experimental procedure section, the plastic and elastic deformed area by nano-indentation was expanded to a depth that corresponds to five times the length of the nano-indenter depth, which was 3 m in this work. Therefore, the indentation hardness data does not involve irradiation-hardening data at the peak damage area where tangled dislocations and many tiny He bubbles formed during He irradiation.” There was no mention of nanoindentation in the methodology section but now the authors are referencing nanoindentation. Also there is no presentation of nanoindentation results. If the authors indent to 300nm depth the indent volume could encompass the irradiation region and they may compare the hardness obtained at that depth with the corresponding unirradiated sample. The difference in hardness between the two values by-passes the size effect experienced at 300nm.
Response 10: Honestly saying, a half of nano-indentation test has been already published in the in this journal in the early 2021. I worried about the duplicate publication at first. As a matter of fact, we succeeded to develop the flash elecrtro-polishing for ion-irradiated vanadium alloy and proceeded the research about the correlation between microstructural change and mechanical property change using nano-indenter, TEM and APT. Then I want to submit a new paper that the microstructural evolution during ion irradiation is a main topic, At first that I prepare the manuscript, I was afraid that the nano-indentation result was not available to use in the paper, but I found it was difficult to use the data without detailed description according to the reviewer’s suggestion. So I write the nano-indentation test in the method and procedure section and I added complemental texts in the result and discussion section.
- Do the authors intend on characterising the helium bubbles introduced or are they purely interested in the tail of the irradiation profile where it is mostly displacement damage?
Response 11: Honestly speaking, I do not have any interest in He bubble formation in this work. The target of this research work is concentrated on the Ti content dependence on Ti(CON) cluster formation induced by irradiation in the middle part of He ion range. Actually He bubble and tangled dislocations formed near He ion range was almost same among all irradiated specimens where He concentration was supersaturated for the excess vacancies produced by irradiation and there is no typical microstructural change among all specimens.
To ignore the damage structure at peak position of He ion range, I added the text in the DBH explanation part in the discussion section as follows; “For this calculation, He bubbles and tangled dislocations at the peak position of He ion range was not considered because it is assumed that the depth area affected by the nano-indentation did not reach to the peak position. The He-irradiated V-4Cr and V-4Cr-0.1(h) alloys showed low irradiation hardening regardless of the existence of highly damaged area with He bubbles and tangled dislocations near ion range at 3.5~3.8mm.”
- Discussion
- The authors need to be more precise in comparing neutron irradiation experiments with the helium irradiation performed. For example, what neutron damage was introduced what alloys were studied. What was the size and number density of the various defects and precipitates and how did it compare with the values measured by helium ion irradiation.
Response 12: I. do not want to compare this experiment data with the previous results about neutron-irradiated V-Cr-Ti alloys directly. To get any insight from the comparison, there is too little data to compare to clarify the essence from both results, Here all I want to say is the speriorit of flash electropolishing for TEM observation in ion-irradiated vanadium alloys through the temperature dependence of cluster formation of irradiated vanadium alloys.
To avoid misunderstanding and duplication, I removed the first paragraph of discussion section.
- In applying the dispersion hardening model to estimate the increase in yield strength the authors ignores the loop density as well as helium bubble density and only uses the precipitate density which is in accurate. The entire section needs to be re-written including contributions from helium and loops.
Response 13: I reconsidered this discussion part including the hardening contribution of dislocation loops. He bubble and tangled dislocation at the peak position of He ion range do not work for the irradiation contribution because the deformation generated by nano-indentation did not read to the peak position. It was apparent from the fact that Ti-bearing V-Cr alloys (V-4Cr and V-4Cr-0.1Ti) showed little irradiation hardening regardless that He bubble and tangled dislocation were formed at the peak position of He ion range. So we ignored the hardening contribution of He bubble and tangled dislocation at the peak position of He ion range from the calculation of irradiation hardening estimation. To include the contribution of dislocation loops for DBH model calculation, the equation for calculation of yield stress increase was changed. We also replotted Figure 8 with the irradiation hardening contribution of dislocation loops.
The track changed part is the following;
“To understand the correlation between microstructural change and mechanical property change in the irradiated materials, the classic dispersed-barrier-hardening (DBH) model was used to evaluate the irradiation hardening that was connected to the microstructural information by applying the density and size of defect variants [28,29]. For a correct statistical estimation of the defect cluster density from the TEM images, the measured density was weighted by using the visibility of defects in a TEM image [19,25]. The tangled dislocation and dislocation loops were negligible because the dislocation density in the midrange of the ion damage profile was low and only Ti(CON) precipitate was available to estimate irradiation hardening. Because one-third of the Ti(CON) should be invisible by the g = [110] reflection in the bright-field image under TEM observation, the Ti(CON) density should be weighted by a factor of 1.5. To obtain the irradiation hardening value from the microstructural changes in the material, a dispersed barrier model was used, and the yield stress increased because of the microstructural changes that were expressed in Equation (1):
∆σ_y= √(∑▒(Mα_i μb√(N_i D_i ))^2 ) (1)
where M is the Taylor factor (3.06 for polycrystals); m is the shear modulus of the matrix (46 GPa); b is the Burgers vector (0.261 nm); Δσy is the increment in yield strength due to Ni number density, Di size of discrete and ai barrier strength of discrete defects of type i such as Ti(CON) precipitates and dislocation loops. The data from the Ti aggregates observed by APT measurement were not used and only the TEM data for the TEM-visible defect clusters were used in this estimation. When C is a conversion parameter from nano-indentation hardness to yield stress to determine the equivalent relationship as ∆σ_(y-total)^Calc≈C∙∆H_it,, according that a value of C was confirmed by (∆H_it)⁄(∆σ_(y-total)^Calc ).. The value of C was determined as 0.09 from other work on self-ion-irradiated stainless steel [30]. The values of M, m, and b are fixed parameters that are specific to a material, and the values of N and D are provided by the measurement of microstructural changes in the irradiated specimens. Previous studies have reported that the value of a for the defect cluster ranges from 0.34 to 0.56 [18,19,25,31], and the barrier strength a for the Ti(CON) precipitate was 0.4 in this work after fitting the data with irradiation hardness changes. The barrier strength a for the dislocation loops was 0.35 in this work For this calculation, it was not considered that He bubbles and tangled dislocations at the peak position of He ion range were not contributed to irradiation hardening in the nano-indentation test because it is assumed that the depth area affected by the deformation due to the nano-indentation did not reach to the peak position. It was apparent from the experimental results that the He-irradiated V-4Cr and V-4Cr-0.1(h) alloys showed low irradiation hardening regardless of the existence of highly damaged area with He bubbles and tangled dislocations near ion range at 3.5~3.8mm. Then the contribution of damage structures at the peak position of He ion range was ignored in this calculation of irradiation hardening estimation by using the DBH model,.
Figure 8 presents a plot of the balance of increase in yield stress in the V–Cr–Ti(h) and V–Cr–Ti alloy after ion irradiation. The measured irradiation hardening data and the estimated yield stress increases are shown by open circles and a bar chart, respectively. A gray and black bar chart shows hardening contribution of Ti(CON) precipitates and dislocation loops, respectively. Most of irradiation hardening is contributed by the Ti(CON) precipitate as shown in Figure 8.”
Round 2
Reviewer 2 Report
Thanks for making the changes suggested the paper is in a good state suitable for publication.